# Supporting Traumatic Grief: A Qualitative Analysis of Helper’s Lived Experience

**DOI:** 10.3390/ijerph192316002

**Published:** 2022-11-30

**Authors:** Diego De Leo, Marta Torres, Andrea Viecelli Giannotti, Josephine Zammarrelli, Laura Tassara, Michela D’Ambros

**Affiliations:** 1Australian Institute for Suicide Research and Prevention, Griffith University, Brisbane, QLD 4122, Australia; 2Slovene Centre for Suicide Research, Primorska University, 6000 Koper, Slovenia; 3De Leo Fund, 35137 Padua, Italy

**Keywords:** traumatic death, helper, helpline, strategies, impact, crisis support

## Abstract

*Background*: Usually managed by trained volunteers, crisis helplines services are of primary importance in community care. However, literature has shown that volunteers may be exposed to a high risk of developing negative psychological consequences in relation to the service performed. Although there are numerous studies dedicated to the use of helplines, evidence regarding the experiences of helpers in the context of traumatic losses is still limited. *Objective*: This study aimed to explore lived experiences of the operators of a crisis line supporting traumatic bereavement. Specifically, it analyzed the psychological impact of interacting with the caller, and the resources and strategies used by the operators themselves. *Methods*: The study considered 35 operators of the helpline provided by the Italian NGO De Leo Fund, which offers psychological support to survivors of traumatic bereavement. The inclusion criteria were: (a) currently being or having been a helper at the De Leo Fund helpline; and (b) having completed at least 10 shifts in the helpline service. Data were collected through an ad hoc questionnaire, filled online. The thematic analysis technique used Atlas.ti software 8. *Results*: Four thematic areas emerged from the transcription of the semi-structured interviews. These were: (a) motivation and expectations; (b) emotional and relational impact of the experience as a helper; (c) strategies adopted by operators; and (d) criticalities and strengths. *Conclusions*: Monitoring helpers’ emotional experiences and the impact of their work overtime emerged in a marked way. The analysis of the strategies and experiences of the operators represents a fundamental factor for the implementation of specific training programs for the management of emergency situations.

## 1. Introduction

Helplines are free services that, through telephone lines, chat or other messaging programs try to provide immediate support to those experiencing a difficult moment [1]. The ease of access to services, their immediate usability and the protection of anonymity qualify crisis lines as fundamental aid in the context of community care, in which the importance of offering wide-ranging and transversal instruments to different population groups appears as paramount [2].

Helplines services are usually run by volunteers, recruited and trained to provide an active and non-judgmental listening service [3,4,5]. The present study was carried out within the helpline service of De Leo Fund, an Italian non-governmental organization (NGO) specialized in providing emotional and psychological support to survivors of traumatic deaths, i.e., people who have lost a loved one in a sudden, unexpected and violent way, such as in case of suicide, murder, road and work accident or natural disaster. The derived bereavement process is defined as ‘traumatic’ because the characteristics of the event that caused death trigger typical reactions of a traumatic experience in those who remain [6,7].

The helpline service of the NGO De Leo Fund has been active since 2007. It is usually provided by a team of volunteer operators, specifically trained to offer empathic and non-judgmental listening, in the context of complete anonymity between telephone caller and the helper. The systematic review by Willems et al. [8] underlines the role of volunteers within crisis lines, in addition to traditional formal care [9]. However, the scientific evidence regarding the figure of the helper in the context of bereavement is still limited, especially for traumatic bereavement.

On the basis of available data, the literature has highlighted a worsening of mental wellbeing in up to 77% of operators, underlining how volunteers (defined as helpers) are exposed to a high risk of developing negative psychological consequences in relation to the service provided [9]. Similarly, factors related to the organization, such as continuous training and supervision and individual characteristics of the operator (e.g., coping strategies and previous experiences) can mediate the negative impact of the service and increase the level of wellbeing of the helpers [8,10,11].

The goal of this research was to explore the experience of the operators of a crisis line in supporting traumatic bereavement, in order to analyze the psychological impact of this type of service, and the resources and strategies used by the operators themselves. The helper, in fact, deals with the emotions and experiences reported by the caller, from which feelings of dissatisfaction may emerge during particularly challenging and/or traumatizing calls [12,13]. Difficulties may emerge with regard to the recruitment of volunteers and the high turnover of the same [11]. This issue is of great interest to associations operating in the sector (including De Leo Fund, which encounters the same problems); however, data appear particularly limited with specific reference to traumatic bereavement, which justify the present investigation.

All this underlines how the psychological well-being of helpers can be jeopardized by the type of activity they carry out: in fact, continuing contact with suffering and despair of callers can affect the psychic health of helpers. With this consideration in mind, what makes this study potentially useful is hearing the voices of helpers, in trying to understand where the difficulties lie in contacting that type of users. The direct experience of those who act in the field may provide valuable information on which direction to take, in order to protect the operators themselves and eventually improve their service. The Executive Board of the NGO De Leo Fund gave ethics clearance to the realization of present investigation among ex-and present operators of the helpline. A team of clinicians was set up to take care of any possible adverse reaction in participants during the investigation.

## 2. Methods

### 2.1. Procedure

Data were collected through the creation of an ad hoc questionnaire (see Appendix A) to be compiled online. The questionnaire consisted of 16 questions, relating to the overall experience of the helper:characteristics of the operator (frequency of service, personal experiences of traumatic bereavement),motivation and expectations of operators in wanting to dedicate themselves to this type of service,experience as a helper (weaknesses, strengths),sphere of the operator’s interpersonal relationships (any changes),emotional impact and strategies used,role of the training received.

### 2.2. Data Analysis

The data of this research were examined through the thematic analysis technique [14] using the Atlas.ti Windows software (version 8.4, by Atlas GmbH). Two researchers (M.T. and M.D.A.) became familiar with the data obtained from the questionnaire. The answers were read several times and first impressions noted. The same researchers generated the initial codes by collecting the resulting salient data for each code. Subsequently the codes were collected in themes and the latter were revised. In the next step, a third researcher (L.T.)—alien to the first part of the study—reviewed the themes and built the definitive themes together with the first two researchers and then labeled them. Once this process was concluded, a final report was produced with the description of the results obtained.

## 3. Results

The participants to the study were recruited from current and former operators of the NGO De Leo Fund. All these individuals received the same specific training on traumatic grief. The inclusion criteria were: (a) currently being or having been a helper of De Leo Fund, and (b) having worked at least 10 shifts in the helpline service. De Leo Fund is operational from Monday to Friday, from 9.00 am to 1.00 pm and from 3.00 pm to 7.00 pm. Therefore, each shift consists of four hours. Each phone call received is recorded and subsequently supervised by a senior professional figure from De Leo Fund (psychologist or psychiatrist). Since its creation, many professionals and volunteers have operated at De Leo Fund’s phones. In all cases, operators were/are represented by trainees or qualified psychologists, and in three cases by medical doctors.

At the time of this study, out all the people that served at De Leo Fund, an email address was available for 60 of them. Thirty-five people subsequently accepted to join the study. Previous operators may not have responded due to the lack of updated contacts or for having provided less than ten shifts or for not wanting to participate. Those who accepted were people of both sexes, with age range from 23 to 64 years, from various regions of Italy. They all had a background in psychology, with at least a three-year degree in psychology. Among them, some were still finishing their studies at the time of the research while others were working as psychologists. A background in psychology could be an asset in dealing with delicate or difficult conversations but probably not enough to protect the operators from the phenomenon of vicarious traumatization. Out of the total sample of 35 participants, 29 individuals were ex-operators (with various time distance from the last service provided), while 6 people are current operators. Majority of participants (20 out of 35 total) carried out more than 5 shifts per month, 5 carried out from 3 to 5 shifts per month, while 10 participants performed from 1 to 2 shifts per month (see Table 1).

Fourteen operators carried out the service for more than 1 year, 15 operators for a period from 6 months to 1 year, and 6 participants for 6 months or less. It was not possible to precisely indicate the number of calls received by each individual operator. Among helpers, 37.1% (13 out of the 35 participants) had suffered a personal experience of traumatic bereavement.

Each participant signed an informed consent to the research, with guarantee of anonymity.

The analysis carried out on the transcripts of the semi-structured interviews allowed the categorization of four thematic areas, described below (Table 2).

### 3.1. Motivation and Expectations

From the results, a link seems to emerge between motivation (what moves towards something) and expectation (what I expect from it); in fact, we can see how people orient their expectations from their motivation. Those who are motivated by a desire to help others expect being able to help. Similarly, those who are motivated by educational reasons, have the expectation to increase their skills.

#### 3.1.1. Motivation and Expectations Related to the Type of User

For 15 operators, the motivation to become a volunteer in a crisis line for traumatic bereavement was guided by the interest in that specific type of user; in other words, the desire to be able to help others through the experience of loss.

From the words of an operator: *“In addition to a personal experience, the possibility of listening to people and their stories lived directly in the first person and being able to support and reassure them in the short term gave me the hope of being able to be useful and helpful in times of need”*.

The questionnaire revealed an internal consistency between this type of motivation and expectations before taking up service: out the 15 operators, 10 expressed the expectation of being able to exert a helping function for users.

#### 3.1.2. Motivation and Expectations Focused on Vocational Training

For 12 operators the reasons were inherent to their training: *“I had the desire to put myself at the service of society, especially those who face suffering or psychological distress. I chose the De Leo Fund because I wanted to learn more about the theme of bereavement and because this NGO seemed reliable to me”*.

Almost all the operators motivated by professional growth expressed the expectation of being able to increase their knowledge and skills (10 operators). *“I was expecting a complex role that required constant training. A role with great responsibility, during which I would have had to be able responding to difficult requests. I also had expectations regarding my professional training, considering this role as a necessary step to start doing this job”*.

#### 3.1.3. Greater Knowledge of One’s Abilities

Five participants expected to acquire new skills, of which four expressed the desire to test themselves on a personal and human level, while one participant wanted to test skills and competences acquired through university courses.

#### 3.1.4. Personal Elements in Choosing to Become a Helper

For the remaining number of participants (8 answers), the motivation arose from personal experiences of loss (7 answers) and suicidal thoughts in the past (1 answer) (Table 3).

### 3.2. Emotional and Relational Impact of the Experience as a Helper

This theme involves several aspects: awareness of the possible emotional impact before starting to serve; the emotions felt during the shift, and any personal changes perceived following the helper activity (Table 4).

#### 3.2.1. Awareness before Starting of the Possible Emotional Impact from Callers

Most operators (27 responses) stated that they considered the possible emotional impact before starting to serve, with reflections on the possible emotional consequences and difficulties by the service and awareness of playing a very delicate role: *“I expected that it would have been a demanding and very impacting service at the emotional level, but it would also have made me growing both as a person and as a professional”.*

#### 3.2.2. Emotions Experienced by the Operators during the Helpline

All participants felt intense emotions during contact with users: frustration (1 response), anger (1 response), distrust (2 responses), inadequacy (2 responses), insecurity (2 responses), fear (5 responses), anxiety (6 answers), helplessness (8 answers), and sadness (8 answers).

Regarding the feeling of impotence, one operator stated: *“I also experience a sense of helplessness and impotence because I don’t know how to respond or what to say when faced with some of the stories I hear. Sometimes I don’t even know how people find the strengths to go on”*.

About half of the operators (14 respondents) also reported feelings of esteem towards the caller (1 response), tenderness (1 response), hope (2 responses), compassion (2 responses) or felt moved (2 responses).

#### 3.2.3. Personal Changes That Have Occurred since the Start of Service within the Helpline

All participants felt a personal change resulting from the helper activity. Most (27 responses) reported personal growth in terms of emotional skills (12 practitioners), increased sensitivity (7 practitioners), human growth (6 practitioners) and greater emotional strength (2 practitioners). *“I think it gave me another way of looking to everyday life and relational experiences, a re-dimensioning of daily or interpersonal problems.”*

For others, changes occurred in the form of both personal and professional growth (8 operators). *“Yes, I think it has changed me entirely, and that my change has certainly influenced my relationships as well. I have perhaps become more sensitive and attentive to this type of experience.”*

### 3.3. Strategies Adopted by the Operators

This thematic area includes “online” strategies, that is, personal techniques for managing feelings and emotions during the call, and “offline” strategies, used to process the experiences and criticalities that emerged after the contact (Table 5).

#### 3.3.1. “Online” Strategies to Manage Feelings and Emotions during the Helpline Service

The operators reported using empathy (13 responses) as a technique able to control and recognize emotions and create a climate of ‘team sharing’: *“As soon as I answered, emotions were very strong; then, when I started listening to the user, the emotions left room to the desire of being useful and to the attempt of helping with everything that was in my strings and that I had learned. This, while empathizing and remaining as aware as possible of my internal states, I concentrated on the callers and their experiences”.*

Most helpers reported the use of active listening (17 responses) as primary strategy to understand the needs of callers, feeding a climate of openness and acceptance. However, several operators reported the need to maintain an emotional detachment in order to maintain their own balance (13 operators): *“I try to listen to the person while we are online, to make the experiences she tells resonate within me in order to be really close to her. However, I maintain some sort of waterproofing, because within the interaction I want the feelings of the other to have absolute priority, not mine”.*

#### 3.3.2. “Offline” Strategies after Contact with Callers

Peer support (20 responses)—expressed in sharing (13 responses) and in group discussions with other operators (7 responses)—is reported as the main strategy for managing and reworking experiences and emotions after the contact with callers: *“The peer support was of fundamental importance, as well as the supervision by the senior psychologists”,* and *“The exchange with the other volunteers makes it possible not to feel the only one experiencing certain situations or difficulties”.*

The importance of supervision (14 responses), individual and in a team, and ongoing training emerged with evidence. Twenty-eight participants clearly expressed how useful is the training for the helper role: *“Very, very much. I would say it is fundamental; without training, I would have been paralyzed in the face of the first contacts or I could have abandoned the internship”.*

### 3.4. Criticalities and Strengths

For the last thematic area, critical elements and strengths, ideas for improvement emerged (Table 6).

#### 3.4.1. Criticalities

In helpers’ experience, criticalities were mainly found in relation to emotional aspects: 8 participants reported difficulty in withstanding the emotional impact, performance anxiety and unpredictability, as it can be seen from these words: *“The problem is always the fear of not being able to listen well; fear of being overwhelmed by the pain of the caller … or performance anxiety”.* Or: *“It is a very complex experience, which also requires—in my opinion—great capacity of questioning yourself. Sometimes the emotions that come from the Other are devastating, and it is not always possible to live up to it”.*

#### 3.4.2. Strengths

All operators reported the cohesion of the team as their strong point (35 responses), expressed in the possibility of freely sharing their experiences: *“I have always appreciated the strong harmony that I have seen and heard in the team, as well as the initial training that I found as not only adequate for the role I would have held but also generally useful for my profession as a psychologist”.* Another operator said: *“The coaching at each shift, my internship mate, but also listening to the experiences of other volunteer operators… all of this transmitted to me interest and skills and ensured that each shift could become an opportunity for growth and discovery of different ways of approaching the callers and the problem of traumatic bereavement in general”.* According to eight operators, the training opportunities offered by the association, such as the practice of role playing, proved to be important strengths.

#### 3.4.3. How to Improve the Helper’s Experience

Six respondents suggested strengthening the training also through meetings with experts (3 responses): *“Make case studies or in any case spend even more time comparing calls received by different helpers and reflect together on different ways of managing those calls”*. Finally, there were suggestions for strengthening peer support by creating more moments of sharing and discussing.

## 4. Discussion

This study analyzed the experience of volunteers operating in a specific helpline for traumatic bereavement, with the aim of exploring the ways in which the dramatic experience of callers might affect helper’s experience, before, during and after the contact with callers. The analysis of each theme, and of the related sub-themes, made it possible to define significant elements of this experience.

Based on the analysis of expectations on the role of operators within De Leo Fund, it emerged that most operators were aware of interfacing with violent and traumatic experiences of loss, thus entering in close contact with a specific type of caller and having to play a complex and delicate role. The helper functions, associated with role expectations, were made explicit by most operators, aware that callers are particularly needy individuals, in a situation of extreme pain and in search of protective closeness.

Most helpers appear as activated by a motivational system oriented towards providing care, i.e., offering care to individuals in a state of extreme fragility and difficulty, requesting comfort and protection This motivational system seems to be especially activated in the presence of traumatized subjects who tend to develop a specular motivational system, oriented towards attachment [15], and aimed at obtaining help and protective closeness from another person identified as appropriate.

This evidence is supported by the theory of intersubjective motivational systems [16,17], which argues that the motivation of helpers is also linked to specific emotional experiences such as compassion and tenderness, understood as emotions typically involved in the offer of care towards a vulnerable person seeking help.

Of further interest are the emotional experiences reported by the operators during the calls. In dealing with traumatically bereaved people, helpers frequently report emotions such as sadness, anxiety, inadequacy, fear, anger, frustration, helplessness, and insecurity. The traumatic experience of the caller can be pervasive and directly associated with the emotional experience of the operators operating in a crisis line. The literature has amply demonstrated that, encountering survivors’ narratives, the operator can suffer a secondary (or vicarious) traumatization [13,18]. The present research has highlighted that, in the context of a helpline for traumatic bereavement, the effect of vicarious traumatization can already be experienced during the telephone contact, through the empathic involvement with the traumatic experiences of the callers.

In general, the results of this study show that the operators have lived and live the experience of being a helper in a positive way. From the thematic analysis it emerged that helpers feel they can orient themselves to the future with a new outlook, enriched by a broader perspective on the meaning of life and human suffering. Empathic capacity, sensitivity, security, self-determination, awareness of pain, respect and value of emotions and relationships appear as increased.

It is conceivable that managing the call during and after the contact leads the helper to somehow integrate the traumatic experience of the callers into their own life experience. At the clinical level, this is what happens when the so-called integration phase of trauma occurs in traumatized patients [19,20]. It is the phase in which individuals give themselves the opportunity to orient to the future with new forces and new energies arising from the relationship in a prospective view. Feeling supported, individuals could reinterpret the traumatic event as an element that strengthens rather than weakens, and eventually gives a new meaning to their life.

These results are interpreted in the light of modern theories on trauma which identify the element of dissociation-integration as a crucial factor in the traumatic experience. Indeed, Van der Hart’s theory of structural dissociation of personality was developed to offer a unifying theory that may explain all trauma-related disorders, from the simpler to the more complex [21,22,23], in which dissociation is defined as an interruption in the usually integrated functions of consciousness, memory, identity or perception of the environment and where integrative failure becomes the element identifying traumatic experiences. Numerous clinicians, in fact, have proposed that post-traumatic stress disorder (PTSD) is a dissociative disorder rather than an anxiety disorder [24].

With reference to adopted strategies, De Leo Fund operators reported that active listening and empathy are the most used techniques with the type of callers they deal with. Active listening is based on the creation of a positive relationship, characterized by four levels of alliance: the suspension of value judgments, participatory listening, verification of correct understanding, and care of the logistics of the communicative context. Specifically, the helpers emphasize the need to keep an active attention on the person, focusing on the contents and emotions of the callers. This strategy is associated with the desire to understand and identify the needs of the caller, accompanied by an attitude of non-judgmental openness while trying to remain calm and stable. Empathy, understood as the ability to “put oneself in the other’s shoes”, has been reported by most operators.

In line with the results that emerged from the exploration of motivations and emotional impact during the calls, it seems reasonable to hypothesize that helpers are inclined to use counselling techniques that can perform a protective function for callers as well as for operators themselves. The focus on the person, typical of the active listening, and the use of empathy as a tool capable of acting as an “emotional filter” with respect to the contents expressed by the caller, are connected to the need for operators to protect themselves from the traumatic content of the communication, in order to result as useful and effective in their helping function.

This research has highlighted that many operators have felt the need to distance themselves emotionally from the content expressed by callers, regardless of the use of empathy as an ability to tune in with the other. The importance of the “emotional detachment” that emerged in this research introduces the need to identify a new listening technique to be adopted in crisis lines in contact with traumatized subjects, in order to prevent phenomena such as vicarious traumatization, compassion fatigue and burnout.

Active listening and empathy, used in the moment of contact with the user, are then replaced by “offline” strategies such as sharing and consulting with the rest of the team (psychologists in training and experienced operators of the Centre), which the operators also define as the most significant strengths of their experience as helpers.

These results may make us reflect on the need to train better and monitor operators in contact with traumatized people—starting from their motivational structure—on at least two fronts: care and cooperation. It is necessary that the exercise of empathy and active listening during the call can be translated into a collaborative and peer approach with the caller, in order to validate and regulate the emotional experiences, and promote the capacity for mentalization. At a later stage, the same approach must be promoted within the team, to contain and elaborate what emerged during the call. In this way, the criticalities connected to the service such as negative experiences, difficulties associated with the management of complex requests and the unpredictability of the type of contacts can be mitigated through the presence of someone to lean on to face the tension and not bring with them the anxieties and negative feelings related to the traumatic contact [25].

These findings may also instruct clinical practice with subjects facing traumatic bereavement by prompting a basic reflection. We need to be aware that bereavement is not a pathology but a tragic event that many people must cope with during their life. In this sense, the bereaved person does not need a “medicalized” approach but an empathic and non-judgmental human support, able to accept the pain. In these terms, the relationship with the help seeker would be structured in a “symmetrical” way, favoring a relationship based on sharing experiences, with the aim of gradually integrating the traumatic experience through the clinician’s ability to tune into the person’s emotional experiences.

## 5. Limitations

The research has some limitations that may have affected the results obtained. The sample turned out to be small. Regarding this, the population taken as reference is part of a very specific category (helpers of a crisis lines in the traumatic bereavement field), contextualized in a specific service (De Leo Fund). The qualitative nature of the study is not meant to claim any representativeness of the totality of operators of De Leo Fund or any other agency performing the same type of activities.

In addition, out of the 60 participants initially contacted, only 35 replied to the questionnaire. Moreover, it was not possible to verify how many calls each individual operator had received. These data could have provided information with respect to length of service not only as a time factor but also in relation to the experience accumulated by each individual helper.

Another relevant element to be denounced is that nearly all participants had a psychological background (out of the 60, three were medical doctors). This variable might have influenced the type of content that emerged from the responses to the questionnaire.

## 6. Conclusions

The complexity of this interesting subject area reinforces the idea that further research is needed, as important information gaps persist [9]. Furthermore, the literature highlights that operators do not always receive the right training before being involved in the task, which is complex and has an impact on the emotional level and psychological well-being [12].

Considering the subjective reports and personal experiences described by those directly involved in the De Leo Fund helpline can help improve the quality of the services dedicated to the people who live a difficult moment of their life due to the loss of a loved one. The promotion of specific training programs [25] and guidelines could also facilitate the acquisition of the resources necessary for the management of emergency situations and act as a protective factor against the negative consequences on the helper’s health. Among the most extreme repercussions, we have mentioned the risk of emotional exhaustion and burnout [10] and secondary traumatization [13,18], the latter yet to be fully explored in the specific area of traumatic bereavement.

Awareness of the consequences of a lack of attention to the helpers’ emotional experiences and the long-term impact of their work appears to be particularly important. In this regard, it could be useful to stimulate discussion between volunteers who work in different helplines. This would make it possible to grasp differences and common aspects between the different experiences, by considering the type of service provided, the type of user and the specific request of the caller. Regarding this study, being De Leo Fund possibly the only NGO in Italy to specifically offer helpline services to support traumatic bereavement, the comparison with international realities could prove to be particularly useful and stimulating.

Future work could also focus on aspects that are still poorly identified, such as the definition of systematized follow-up phases to ensure continuous and effective monitoring of the helper’s experience.

## Figures and Tables

**Table 1 ijerph-19-16002-t001:** Characteristics of participants (*n* = 35).

Active and No Longer Active Operators		Number of Monthy Shifts			Lenght of Service			Personal Experience of Traumatic Bereavement	
Active	No longer active	More than 5	From 3 to 5	From 1 to 2	More than 1 year	From 6 months to 1 year	Less than 6 months	Yes	No
6	29	20	5	10	14	15	6	13	22

**Table 2 ijerph-19-16002-t002:** Themes and sub-themes obtained from the thematic analysis.

Theme	Sub-Theme
Motivation and Expectations	Users
	Training
	Greater knowledge of one’s abilities
	Personal elements
Emotional and relational impact of the experience as a helper	Awareness before starting of the possible emotional impact with callers
	Emotions experienced by the operators during the shift
	Personal changes
Strategies adopted by operators	Online strategies to manage feelings and emotions during the shift
	Offline strategies after contact with callers
Criticalities and strengths	Criticalities
	Strengths
	How can helper experience be improved

**Table 3 ijerph-19-16002-t003:** Theme 1: “Motivation and Expectations”, sub-themes, codes emerged and related frequencies.

Sub-Theme	Codes	Frequencies
Users	Importance of interfacing with users	1
	Willingness to help others	14
Training	Thesis and internship	2
	Reliability of the NGO	1
	Willingness to increase one’s skills	9
Greater knowledge of one’s abilities	Test yourself	4
	Test your skills	1
Personal elements	Personal experiences of loss	7
	Suicidal thoughts in the past	1

**Table 4 ijerph-19-16002-t004:** Theme 2: “Emotional and relational impact of the helper experience”, sub-themes, codes that emerged and related frequencies.

Sub-Theme	Codes	Frequencies
Before starting, awareness of the possible emotional impact from callers	They considered it	27
	They didn’t consider it	2
	Not relevant	1
Emotions experienced by the operators during the shift	Frustration	1
	Anger	1
	Distrust	2
	Inadequacy	2
	Insecurity	2
	Fear	5
	Anxiety	6
	Impotence	8
	Sadness	8
	Esteem	1
	Tenderness	1
	Felt moved	2
	Hope	2
	Compassion	2
Personal changes	Personal growth	27
	Professional growth	8

**Table 5 ijerph-19-16002-t005:** Theme 3: “Strategies adopted by operators”, sub-themes, codes that emerged and related frequencies.

Sub-Theme	Codes	Frequencies
“Online” strategies to manage feelings and emotions during the helpline service	Empathy	13
	Active listening	17
	Emotional detachment	13
“Offline” strategies after contact with users	Peer support	20
	Supervision	14
	Training	28

**Table 6 ijerph-19-16002-t006:** Theme 4: “Criticalities and strengths”, sub-themes, codes that emerged and related frequencies.

Sub-Theme	Codes	Frequencies
Criticalities	Emotional sphere	8
	Difficulty in handling complex requests	1
	Unpredictability	2
Strengths	Team cohesion	35
	NGO’s tools	8
How you can improve the experience of helper	More training	9
	Meetings with experts	3
	Strengthening Peer Support	18

## Data Availability

All data (interviews transcripts) are available on reasonable request.

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
