# Peer review of "Supporting Traumatic Grief: A Qualitative Analysis of Helper’s Lived Experience"

_ijerph, 2022, doi:10.3390/ijerph192316002_

Round 1

Reviewer 1 Report

Abstract:

Some contents in the Method section are redundant. Please shorten it.

Also, the number of participants was neglected in the abstract.

In the introduction, the research niches/ gaps didn’t seem to be obvious. Please emphasise the intentions of why this study is needed. What significant issues you were going to resolve? Please clearly demonstrate them.

The sample size seemed to be small. Based on what rationales, you believed that this sample size is sufficient enough to support your findings? Please give support for the sample size.

In the Methods section, the characteristics of the participants were stated. The characteristics of the participants should be stated in the Results section.

I recommend that the questionnaire could be placed in the supplementary section instead of Table 2. If you would like to show some questions in the questionnaire, you can put several questions as an example.

Did you get any ethical approval for this study? If not, this study cannot be conducted.

Author Response

Point 1: Abstract: Some contents in the Method section are redundant. Please shorten it.

Also, the number of participants was neglected in the abstract.

Response 1: In the abstract, the methods section has been modified and the number of participants has been added.

Point 2: In the introduction, the research niches/ gaps didn’t seem to be obvious. Please emphasise the intentions of why this study is needed. What significant issues you were going to resolve? Please clearly demonstrate them.

Response 2: The reasons for the study were explained and described in the introduction. The researchers conducted the study to demonstrate the decrease in psychological well-being by crisis line caregivers, particularly for traumatic bereavement. It was necessary to deepen the subject and conduct research also due to the scarce literature on the subject. Investigating the direct experience of the helpers made it possible to understand directly from them what the difficulties of their work are and what can be done to improve their service.

Point 3: The sample size seemed to be small. Based on what rationales, you believed that this sample size is sufficient enough to support your findings? Please give support for the sample size.

Response 3: The study was conducted within a very specific population of subjects, these are helpers of crisis lines for traumatic bereavement. How highly specific is the reality within which the participants were recruited, namely the De Leo Fund, the only Italian company in the area that specifically deals with survivors of traumatic mourning. In addition, the qualitative analysis carried out justifies such a small sample. This type of analysis allows us to grasp the nuances of the survey context in order to be able to compare it with different realities. In addition, the results obtained from this study proved to be representative and in accordance with the currently existing literature with respect to the experience of a helper within a crisis line.

Point 4: In the Methods section, the characteristics of the participants were stated. The characteristics of the participants should be stated in the Results section.

Response 4: The characteristics of the participants have been moved to the "Results" section.

 Point 5: I recommend that the questionnaire could be placed in the supplementary section instead of Table 2. If you would like to show some questions in the questionnaire, you can put several questions as an example.

Response 5: The questionnaire has been included in the supplementary material.

Point 6: Did you get any ethical approval for this study? If not, this study cannot be conducted.

Response 6: The study was not conducted in a public context. The De Leo Fund is an NGO and as such provides within its organization the presence of a committee that deals with the ethical evaluation of the work we carry out. Furthermore, we offer free support services to our users in case of psychological distress. All the work we carry out takes into account the protection of the psychological well-being of the participants.

Reviewer 2 Report

This paper explored the experiences of the operators of a crisis line in supporting traumatic bereavement, in order to analyze the psychological impact of this type of service, and the resources and strategies used by the operators themselves. This research could help improve the mental health and quality of services for this specific group of people. I personally like this article very much.  However, before publishing, I feel the author needs to consider the following advice:

1. This article is so applied that it downplays the dialogue with relevant theories. I would like to see the author adopt relevant theories to interpret the results in the discussion section.

2. While the author studied a special group such as crisis line helpers, psychologists often encounter similar problems. What is the enlightenment and reference significance of the research on the psychological health and services of psychiatrists? The author can explore further in the discussion section.

3. It is obvious that the crisis line service helpers’ own resource endowment, such as knowledge level, age and working experience, will have a great impact on their mental health and service quality. Therefore, the author's choice of study sample may be very important. In order to increase the representativeness of samples, it is necessary to further describe the characteristics of samples.

In addition, some details need to be noted:

1. According to the mdpi template, each section (Introduction, Methods, Results, etc.) and its corresponding subsection (Participants, Procedure, Data analysis, etc.) should be given serial numbers.

2. In the Motivation and Expectations section, give relevant subsection if you can. The purpose of this is to be consistent with the following statement.

3. Refer to the template of mdpi for the table.

Author Response

Point 1: English language and style are fine/minor spell check required.

Response 1: The text has been changed to improve the English language.

Point 2: This article is so applied that it downplays the dialogue with relevant theories. I would like to see the author adopt relevant theories to interpret the results in the discussion section.

Response 2:  For the interpretation of the results, in the "discussion" section, the main reference theories have been added.

Point 3: While the author studied a special group such as crisis line helpers, psychologists often encounter similar problems. What is the enlightenment and reference significance of the research on the psychological health and services of psychiatrists? The author can explore further in the discussion section.

Response 3:  In the concluding part of the "discussion" section, some reflections have been added on the basis of the results that emerged from our study. In particular, we have added some considerations on how to carry the techniques and strategies adopted by helpers with bereaved callers into a clinical setting, favoring the identification of a possible psychological counseling model for people who have suffered a traumatic bereavement.

Point 4: It is obvious that the crisis line service helpers’ own resource endowment, such as knowledge level, age and working experience, will have a great impact on their mental health and service quality. Therefore, the author's choice of study sample may be very important. In order to increase the representativeness of samples, it is necessary to further describe the characteristics of samples.

Response 4: The characteristics of the study participants were further explored: the geographical area of origin and the academic and professional profile were mentioned. In the latter case, it was shown that the psychological training of the volunteers allowed the helpers to come into greater contact with the painful and difficult feelings of the survivors. Each trait of the participants was mentioned ensuring anonymity.

Point 5: According to the mdpi template, each section (Introduction, Methods, Results, etc.) and its corresponding subsection (Participants, Procedure, Data analysis, etc.) should be given serial numbers.

Response 5: Each section of the article has been numbered following the mdpi template.

Point 6: In the Motivation and Expectations section, give relevant subsection if you can. The purpose of this is to be consistent with the following statement.

Response 6: Sub-sections have been added in the section "Motivation and Expectations" to maintain consistency with Tables 2 and 3.

Point 7: Refer to the template of mdpi for the table.

Response 7: the tables have been modified following the mdpi template.

Round 2

Reviewer 1 Report

This manuscript has been improved accordingly and thus I have no concern for it.